# Examining the Gambling Behavior of University Students: A Cross-Sectional Survey Applying the Multi-Theory Model (MTM) of Health Behavior Change in a Single Institution

**DOI:** 10.3390/healthcare11152151

**Published:** 2023-07-28

**Authors:** Sidath Kapukotuwa, Laurencia Bonsu, Anita Chatterjee, Miguel Fudolig, Manoj Sharma

**Affiliations:** 1Department of Social & Behavioral Health, School of Public Health, University of Nevada, Las Vegas, NV 89119, USA; bonsul1@unlv.nevada.edu (L.B.); chatta1@unlv.nevada.edu (A.C.); manoj.sharma@unlv.edu (M.S.); 2Department of Epidemiology and Biostatistics, School of Public Health, University of Nevada, Las Vegas, NV 89119, USA; miguel.fudolig@unlv.edu; 3Department of Internal Medicine, Kirk Kerkorian School of Medicine at UNLV, Las Vegas, NV 89106, USA

**Keywords:** behavior, theory, addiction, gambling, college students, university students, young adults, betting, gaming

## Abstract

Gambling among college students can start as a pastime activity. However, this pastime can lead to problem gambling and pathological gambling. This cross-sectional study aimed to identify and explain the initiation and sustenance of quitting gambling among university students who had participated in gambling during the past month and those who had not using a novel fourth-generation multi-theory model (MTM) of health behavior change. Data were collected from a sample of 1474 university students at a large southwestern university in the U.S. between January 2023 and February 2023, utilizing a validated 39-item survey. The statistical analyses employed in this study encompassed descriptive statistics, independent samples t-tests, and hierarchical regression modeling. Among students who had engaged in gambling in the past month, the constructs of participatory dialogue (β = 0.052; *p* < 0.05), behavioral confidence (β = 0.073; *p* < 0.0001), changes in the physical environment (β = 0.040; *p* = 0.0137), and demographic variables accounted for 27.7% of the variance in the likelihood of initiating the behavior change. Furthermore, the constructs of emotional transformation (β = 0.104; *p* = 0.0003) and practice for change (β = 0.060; *p* = 0.0368), and demographic variables accounted for 22.6% of the variance in the likelihood of sustaining quitting gambling behavior. The Multi-Theory Model (MTM) can be employed to design interventions aimed at reducing problem gambling among college students.

## 1. Introduction

The years spent in higher education mark a significant turning point in an individual’s life, one that bridges the gap between childhood and adulthood. Problem behaviors may occur alongside the considerable rise in freedom and leisure options that characterize this stage of emerging adulthood [1]. In addition to alcohol and drug use, college students are exposed to gambling on campuses and in the college community. One pastime that can quickly become problematic is gambling, especially considering the explosive growth in the industry over the previous two decades among people of all age groups [1]. The importance of gambling as a public health concern has been emphasized more and more, recently [2,3,4]. Drug abuse, physical and sexual violence, and mental health problems, including anxiety and depression, are negative outcomes associated with gambling [2].

Legal gambling stations are prevalent throughout the southwestern region of the United States. Among various demographics, the problem of gambling poses a significant concern, particularly among college students. This group possesses the necessary resources, proximity, leisure time, and inclination to engage in various gambling options, including online gambling, lotteries, fantasy sports leagues, and more [5,6]. Research has indicated that around 75% of college students have engaged in legal or illegal gambling within the past year [6,7]. Recent studies have shown that the male gender, depression, low parental guidance or monitoring levels, previous winnings, and attention deficit hyperactivity disorder (ADHD) are antecedents of problem gambling [8,9]. The prevalence of gambling has increased due to the availability and legalization of all forms of gambling [10]. College students may have a higher propensity than adults to develop gambling problems, particularly if they have a fondness for sports or engage in sports betting [11]. Many students gamble to “have fun,” “mingle with friends,” and “present as the cool kids.” However, significant numbers of students engage in gambling or betting to the extent that their actions are consistent with problem gambling or pathological gambling, a mental disease [1]. The estimated lifetime prevalence of problem gambling among college students is approximately 5% [6].

Gambling among college students can lead to issues such as low grades, suicide, missed classes, physical violence, binge drinking, and mental health problems [12]. Problem gambling is characterized by uncontrollable, destructive, and compulsive gambling with significant deleterious personal, social, financial, psychological, vocational, and academic consequences among college students [8,13]. Problem gamblers have little behavioral control and are frequently preoccupied with gambling-related thoughts, which may later lead to pathological gambling [14]. Pathological gambling is a destructive and compulsive behavior that is defined as the inability to regulate the urge to gamble [6].

Given the enduring adverse effects of problem gambling on college students, it is crucial to comprehend and elucidate the gambling behavior exhibited by this population [11]. Numerous studies have employed the theory of planned behavior to explain behavioral intention and problem gambling behavior, which predicts individuals’ behavioral intentions and perceived behavioral control. By contrast, attitudes towards behavior, subjective norms, and perceived control over the behavior significantly influence individuals’ intentions [11,15,16].

This study aimed to explore the correlates of problem gambling behavior among college students using the multi-theory model (MTM) of health behavior change [17]. The unique, fourth-generation MTM examines university students’ gambling behavior. The multi-theory model for health behavior change was developed by drawing upon the collective insights and knowledge of multiple behavioral theorists [17]. The MTM is divided into two components: initiation and sustenance. The concept of initiating behavioral change is synonymous with adopting a one-time behavior. The sustenance, also known as continuation, of a health behavior refers to the long-term performance of that behavior throughout one’s lifetime. It is important to make this differentiation between initiation and sustenance of behavior because the constructs that influence these two aspects are distinct from each other. Existing behavior theories often fail to differentiate between them, leading to limited predictive power when these theories are put into practice [17].

The initiation can be divided into three constructs: (1) Participatory dialogue is based on Freire’s model [18] of adult education. Participatory dialogue focuses on discussing the pros and cons of health behavior changes, similar to the transtheoretical model [19] and health belief model [20]. However, it differs by being a two-way communication process, encouraging mutual exploration. (2) Behavioral confidence (e.g., an individual’s level of assurance in engaging in behavioral change) is derived from Bandura’s self-efficacy [21] and Ajzen’s perceived behavioral control [22], and (3) changes in the physical environment (e.g., individual changes in their physical environment regarding the availability and accessibility of necessary resources) are derived from various sources, including Bandura’s concept of the environment [21], Prochaska’s construct of environmental reevaluation [19], and environmental factors in Fishbein’s integrative model [23]. Sustenance can also be divided into three constructs: (1) Emotional transformation (e.g., changes in a person’s feelings towards behavior change) stems from the self-motivation construct of emotional intelligence theory [24]. (2) Practice for change (e.g., the individual’s capacity to overcome obstacles like losing gambling friends or experiencing bullying because they decided to quit gambling) is derived from Freire’s adult education model’s praxis [18], and (3) changes in the social environment (e.g., the ability to sustain one’s behavioral change through receiving social support from family and friends) are derived from constructs like the environment [21], helping relationships [19], and social support [25], among others.

The MTM has been widely applied to assess various health behaviors such as physical activity, teeth brushing, and COVID-19 vaccine hesitancy [26,27,28], to name a few. Furthermore, MTM has been used to study other addictive behaviors such as vaping, smoking, and substance use [29,30,31]. The combined predictive effect of all MTM constructs on vaping quitting initiation was found to be 41.7%, while for sustenance it was 36.6% [29]. The smoke cessation study was a qualitative study and the interview tool used in the study was guided by the MTM, allowing for a more comprehensive assessment of the personal and socio-environmental factors influencing smoking cessation [30]. In a stepwise multiple regression analysis predicting the initiation and sustenance of intentional substance use cessation, the model accounted for a total of 34.2% of the variance [31]. The findings of these studies demonstrate the practicality and relevance of using the MTM framework in identifying and examining addictive behaviors. However, MTM has not been used to study gambling behavior until now. These questions guided the study; to what extent do the correlates based on MTM explain the initiation of quitting gambling behavior among university students while controlling for demographic covariates? To what extent do the correlates based on MTM explain the maintenance of quitting gambling behavior among university students while controlling for demographic covariates? A secondary aim of the study was to validate a tool for measuring gambling behavior based on MTM. The findings of this study can inform the development of health promotion interventions that effectively encourage the target population to initiate and maintain healthy gambling behavior.

## 2. Materials and Methods

### 2.1. Study Design and Study Participants

This study employed a cross-sectional design, with data collection conducted between 25 January 2023 and 17 February 2023. The study targeted enrolled students at a large southwestern university in the United States. Eligible participants included students aged 18 years or older who proficient in English and who had provided their consent to participate.

### 2.2. Ethical Considerations

On 18 November 2022, the study received an exemption from the institutional review board of the university (Protocol # UNLV-2022-511). Participants provided their consent by voluntarily agreeing to participate in the study. The informed consent form provided participants with comprehensive information regarding the study’s objectives, significance, and associated risks, and the option to withdraw their participation at any time. Participants who selected the “Agree” option were directed to proceed to the survey. No personal data such as names or email addresses were collected.

### 2.3. Recruitment and Data Collection

The participants were initially recruited through announcements in the university’s student newsletters. Then, student directory information was obtained from the registrar’s office, and students were directly contacted through their university emails. Complete details about the study and an anonymous link to the survey were provided through these direct emails. Once participants clicked the link, they were directed to the Qualtrics (Provo, UT, USA) online survey. Participants could enter a draw for USD 20 Visa gift cards. Twenty-five randomly selected participants had the opportunity to receive one gift card as an incentive for participation. To preserve the anonymity of participants, we provided a separate survey link to enter the gift card draw after the completion of the cross-section study survey. Participants were asked whether they were interested in participating in the draw for gift cards. If they selected “Yes,” they were directed to the gift card drawing survey, where we asked them to provide their email address for contact purposes. The “Prevent multiple submissions” option in Qualtrics was switched on to prevent participants from taking the surveys multiple times. The bot detection option was switched on to identify whether bots were taking the surveys, and “RelevantID” was switched on to detect fraudulent responses.

### 2.4. Survey Instrument

The instrument comprised a 39-item survey designed by the theory’s originator. The first question determined whether the students had participated in gambling within the last 30 days. The following nine questions were used to collect the participants’ demographic data. Those were the type of gambling, if they considered their gambling to be problematic, and their gender, age, race/ethnicity, student status (freshmen, sophomore, etc.), GPA, living condition (on or off campus), and working status. The remaining 29 questions assessed the constructs of the multi-theory model (MTM) for both initiation and sustenance models.

Participatory dialogue was measured by perceived advantages and disadvantages. Each was measured by five items. Each item was measured on a five-point scale (0 = never to 4 = very often). The range of scores for perceived advantage and disadvantage was 0 to 20, and the range of scores for the participatory dialogue was −20 to 20 (perceived advantage–perceived disadvantage). Behavior confidence was measured by five items. Similar to perceived advantages and disadvantages, behavior confidence items were measured with five-point scales (0 = not at all sure to 4 = completely sure). The range of scores for behavioral confidence was 0 to 20. The remaining four constructs were measured with three items each. These were also measured with five-point scales (0 = not at all sure to 4 = completely sure). The possible range of scores for each construct was 0 to 12. The overall initial score was measured by one item, which had a five-point scale (0 = not at all likely to 4 = completely likely) with a possible range of scores of 0 to 4. The instrument used a similar method to measure the overall sustenance score. The Flesch reading ease was 71.9 for the instrument, and the Flesch–Kincaid Grade Level was 4.6.

### 2.5. Statistical Analyses

All data were analyzed using SAS version 9.4 (SAS Institute Inc., Copyright © 2016 SAS Institute Inc. SAS and all other SAS Institute Inc. product or service names are registered trademarks or trademarks of SAS Institute Inc., Cary, NC, USA.) and R Statistical Software version 4.3.0 (R Core Team (2021). R: A language and environment for statistical computing. R Foundation for Statistical Computing, Vienna, Austria.). A first-order, multi-factor model was used in the confirmatory factor analysis for initiation and sustenance models. The confirmatory factor analysis (CFA) for construct validation was performed using the R package lavaan [32]. We used Weighted Least Squares with Mean and Variance adjustments (WLSMV) designed for ordinal data [33] in implementing the CFA. The robust estimates of the comparative fit index (CFI), root mean square error of approximation (RMSEA), and the standardized root mean square residual (SRMR) were used to diagnose the fit for the presented model. We used the following cutoff criteria recommended by Hu and Bentler (1999) [34] to assess the acceptable fit: CFI values above 0.95, RMSEA values below 0.06, and SRMR values below 0.08. The internal consistency of the subscales and the entire scale was tested using Cronbach’s alpha values, using 0.70 as the lower threshold for acceptable values. Convergent validity was measured using the average variance extracted (AVE). The AVE values for each subscale should be at least 0.5 to establish convergent validity. The model’s reliability was measured using the McDonald’s omega values using a lower threshold of 0.7. Descriptive statistics for continuous variables were presented using mean and standard deviation, while categorical variables were summarized using frequencies and percentages. Our dependent variables were the likelihood of intention and sustenance of gambling behavior among university students. The independent variables in this study were the MTM constructs, while gambling type, age, sex, race/ethnicity, class, and GPA served as covariates. An independent samples t-test was used to compare the mean differences in MTM constructs between individuals who had not engaged in gambling in the past month and those who had participated in gambling during the past month. Correlations among the independent variables were tested using Pearson’s correlations test. Hierarchical multiple regression analysis was employed to predict the likelihood of initiating and sustaining quitting gambling behavior. Model assumptions, independence of observations, linearity, normality, and equal variance were tested using Durbin–Watson statistics, partial regression plots, the Shapiro–Wilk test, and the White test, respectively. The multicollinearity of the final model was tested using the variance inflation factor (VIF). The significant level was set to 0.05 for all statistical analyses.

## 3. Results

### 3.1. Sample Characteristics

A total of 1474 valid responses were collected. Most of the participants were female students (64.72%, n = 954), and there were 497 male students (33.72%) and 23 students (1.56%) who chose the “Other” option for gender (Table 1). The average age was 25.3 years, with a standard deviation of 7.7 years. There were 595 (40.37%) White, 326 (22.12%) Hispanic, and 236 (16.01%) Asian American students. The remaining 317 (20.51%) students belonged to multi-racial, Black, Native American, or other race/ethnicity groups (Table 1). The majority of participants were undergraduate students (68.92%, n = 1016), most of the students (61.71%, n = 910) reported having a GPA of 3.50 or above, and 912 (61.87%) students reported not participating in gambling for the past month (Table 1).

### 3.2. Reliability Analysis, Construct Validity, and Fit Diagnostics

We established the construct validity for initiation and sustenance models by assessing internal consistency through Cronbach’s alpha, reliability by using McDonald’s omega, and convergent validity by using the average variance extracted (AVE). All calculated values are listed in Table 2. All scales and subscales for initiation and sustenance models had Cronbach’s alpha values above 0.7, which established the instrument’s internal consistency for both models. The McDonald’s omega values for all constructs in both models were at least 0.70, which met the criteria for reliability. The convergent validity of the models was also confirmed by the AVE values, which were above the 0.5 threshold constructs, with recorded values from 0.50 to 0.80.

The fit diagnostics of the confirmatory factor analysis for the initiation model used for construct validation show that the model fit the data we collected. We recorded a robust CFI estimate of 0.95, RMSEA of 0.067, and SRMR of 0.06. The factor loadings for the initiation model ranged from 0.563 to 0.985, implying that the items were highly correlated with the constructs. All these indices fell within the acceptable range of values to establish a good fit, as stated in Section 2.5. The sustenance model also fit the data we collected, with respective CFI, RMSEA, and SRMR estimates of 0.997, 0.023, and 0.025. Like the initiation model, the sustenance model also yielded high factor loadings ranging from 0.659 to 0.909. These findings supported the construct validity of the instrument.

### 3.3. Characteristics of Study Variables and Inferential Statistics

The mean scores differences in the initiation and sustenance of those who had not participated in gambling during the past month and those who participated in gambling were significant (Table 3). The mean initiation score was significantly higher among individuals who had not engaged in gambling during the past month (M = 3.79, SD = 0.70) compared to those who participated in gambling (M = 2.95, SD = 1.37), with a statistically significant mean difference of M = 0.84, 95% CI [0.73, 0.95], *p* < 0.0001 (Table 3).

Likewise, the mean score for sustenance was significantly higher (M = 3.17, SD = 1.18) among individuals who had not participated in gambling during the past month compared to those who had (M = 1.78, SD = 1.48), with a statistically significant mean difference of M = 1.39, 95% CI [1.25, 1.53], *p* < 0.0001 (Table 3).

To assess the impact of MTM constructs on initiation likelihood, hierarchical multiple regression was conducted by sequentially adding the MTM constructs to the demographic variables. The analysis aimed to determine whether the inclusion of the constructs improved the prediction of initiation (Table 4). The full model (Model 4) for those who had not participated in gambling during the past month was statistically significant: R^2^ = 0.165, F(13, 898) = 13.63, *p* < 0.0001; adjusted R^2^ = 0.153 (Table 4). Incorporating participatory dialogue into the prediction of initiation (Model 2) resulted in a statistically significant increase in R^2^ of 0.059. Furthermore, including behavioral confidence in the prediction of initiation (Model 3) led to a statistically significant increase in R^2^ (of 0.072).

In the hierarchical regression analysis of the sustenance model for individuals who had not participated in gambling during the past month, the full model (Model 4) yielded a statistically significant result, with an R^2^ of 0.306, F(13, 898) = 30.47, *p* < 0.0001 and an adjusted R^2^ of 0.296 (Table 5). Incorporating emotional transformation into the prediction of sustenance (Model 2) resulted in a statistically significant increase in R^2^ (of 0.171). Additionally, including practice for change in the prediction of sustenance (Model 3) led to a statistically significant increase in R^2^ (of 0.028).

The full model (Model 4) for those who had participated in gambling during the past month was statistically significant: R^2^ = 0.294, F(13, 548) = 17.56, *p* < 0.0001; adjusted R^2^ = 0.277 (Table 6). Including participatory dialogue in the prediction of initiation (Model 2) resulted in a statistically significant increase in R^2^, of 0.098 (see Table 6). Furthermore, incorporating behavioral confidence into the prediction of initiation (Model 3) led to a statistically significant increase in R^2^, of 0.084. After controlling for the other predictors, when the participatory dialogue was increased by 1%, the likelihood to practice initiation significantly increased by 0.052 (95% CI = 0.035, 0.068; *p*-value < 0.0001) on average. Similarly, after controlling for the other predictors, when the behavioral confidence was increased by 1%, the likelihood to practice initiation significantly increased by 0.073 (95% CI = 0.049, 0.097; *p*-value < 0.0001) on average.

In the hierarchical regression analysis of the sustenance model for individuals who had participated in gambling during the past month, the full model (Model 4) yielded a statistically significant result, with an R^2^ of 0.244, F(13, 548) = 13.60, *p* < 0.0001, and an adjusted R^2^ of 0.226 (see Table 7). Additionally, incorporating emotional transformation into the prediction of sustenance (Model 2) resulted in a statistically significant increase in R^2^, of 0.067. After controlling for the other predictors, when the emotional transformation was increased by 1%, the likelihood of continuing to abstain from gambling significantly increased, by 0.104 (95% CI = 0.048, 0.160; *p*-value = 0.0003) on average. Similarly, after controlling for the other predictors, when the practice for change increased by 1%, the likelihood of continuing to abstain from gambling significantly increased by 0.060 (95% CI = 0.004, 0.116; *p*-value = 0.0368) on average.

## 4. Discussion

Our study employed a novel fourth-generation behavior theory, the multi-theory model (MTM) of health behavior change, to comprehensively explain gambling behavior among university students. Gambling is legal in most of the states in the United States. However, only 38.13% of students reported participating in gambling during the past month. This study shows that despite gambling being legal and students having easy access to gambling, 61.87% on the campus studied had chosen not to participate in gambling during the past month. In our sample, 4.34% of the students reported problematic gambling. The worldwide percentage of problem gambling among college students is 10.23%, which is higher than that in our sample [6]. Still, it can be appreciated that a significant number of university students are engaging in gambling, thus necessitating the need for designing interventions to curb gambling and problematic gambling.

The study revealed that 27.7% of the likelihood of practice initiation variance could be explained by the final model for those who participated in gambling during the past month, and all three initiation constructs were significant predictors. The study findings demonstrated that among those who participated in gambling during the past month, approximately 18.9% of the variance in the likelihood to practice initiation was significantly predicted by all three factors—participatory dialogue, which assesses the advantages and disadvantages of gambling behavior, behavioral confidence, which represents the sureness in abstaining from gambling despite potential barriers, and changes in the physical environment. Those who participated in gambling saw more disadvantages in not participating in gambling than those who did not participate in gambling. Based on the observed results, it is recommended that individual-level behavior change interventions targeting gambling behavior among university students should focus on incorporating the constructs of participatory dialogue, behavioral confidence, and changes in the physical environment. All three constructs were found to be significant predictors of the likelihood of practice initiation among those who had participated in gambling during the past month. There is evidence from other studies on MTM that these three constructs are important for behavior change [26,29,35]. While this is the first study of MTM with gambling behavior, in an MTM study to understand quitting vaping behavior [29], all three constructs of initiation were statistically significant and accounted for 42% of the variance. Similarly, for an MTM-based substance use cessation study, participatory dialogue and behavioral confidence were statistically significant and accounted for 34% of the variance [31]. Likewise, in another MTM-based binge drinking reduction study, behavioral confidence and changes in the physical environment were found to be statistically significant key determinants [36].

The variance in the likelihood of continuing to abstain from gambling for those who participated in gambling during the past month was 22.6%. The study findings demonstrated that among those who participated in gambling during the past month, approximately 7.5% of the variance in the likelihood to sustain quitting gambling was significantly predicted by emotional transformation and practice for change. The study results revealed the significance of emotional transformation, which involves converting feelings into goals of abstaining from gambling in promoting abstaining behavior among university students. Additionally, the contribution of practice for change, which focuses on creating a habit of abstaining from gambling and integrating it into one’s way of life, was found to be statistically significant in predicting the likelihood of continuing to abstain from gambling. These findings highlight the importance of incorporating emotional transformation and practice for change in interventions aimed at promoting long-term abstinence from gambling among university students. The findings are also supported by data from those who did not participate in gambling during the past month. Based on these results, individual-level behavior change interventions for gambling behavior among college and university students should utilize constructs of emotional transformation and practice for change. There is evidence from other studies on MTM that these two constructs are important for behavior change [35,37,38]. While this is the first study of MTM with gambling behavior, an MTM study of related problematic behavior of substance use cessation found practice for change and changes in the social environment to be statistically significant and account for 33% of the variance [31]. Likewise, in another problem behavior, binge drinking reduction among college students, practice for change was found to be a significant construct [36]. In this study, the construct of changes in the social environment was not found to be significant either for those who did not participate in gambling or for those who did. This could be due to gambling behavior being a personal behavior driven by personal choice, and the role of others perhaps not being that important. Future studies need to explore this construct further.

### 4.1. Implications for Practice

The results from this study can be used for designing interventions for college and university students to help them quit gambling or participate in responsible gambling. The finding shows that more than one third of college students have participated in gambling during the past month. To prevent these students from falling victim to problem gambling or pathological gambling, designing individual-level behavioral change interventions is necessary when targeting college and university students. Such educational interventions can be delivered at student wellness centers on campuses. There can be different modalities for such interventions, such as face-to-face educational interventions, asynchronous web-based instruction, remote synchronous classes, social media, and apps. A randomized control trial (RCT) design can be utilized to establish the efficacy of such interventions, where MTM-based interventions can be compared with another theory-based or knowledge-based intervention.

MTM-based interventions have previously been used to target college and university students for various health behavior changes [27,28,35,36,37,38,39,40,41,42]. Participatory dialogue can be built by emphasizing the advantages of quitting gambling, such as lower stress, better health, saving money, better academic performance, less anxiety, etc. At the same time, the disadvantages of quitting gambling, such as less enjoyment, recurrence of cravings, missing feeling a high, and so on, need to be refuted through discussion or other educational tools. By emphasizing the assessment of the advantages and disadvantages of gambling behavior and promoting a strong sense of confidence in abstaining from gambling, interventions can effectively address and modify gambling behaviors among university students [17]. Behavioral confidence can be built through role-playing activities, photovoice, journaling, and introducing students to new hobbies [17]. Changes in the physical environment can be fostered by persuading students to stay away from gambling sites [17].

When looking at constructs of sustenance, emotional transformation is more prominent. To build emotional transformation, converting emotions, especially negative ones, into goals, overcoming self-doubt, and self-motivation for behavior change are vital for sustaining quitting gambling [17]. These can be built through educational programs. Self-awareness of one’s feelings is the first step in building this construct [24]. Once those feelings have been identified, the next step is to create goals. The goal of a gambling behavior change intervention should be that when an individual feels a craving for gambling, they should divert this feeling and initiate another hobby, and participate in it until it feels natural and they no longer have a craving for gambling. Likewise, to influence practice for change, self-monitoring gambling behavior, overcoming barriers, and having contingency plans in relapse are vital for quitting gambling [17]. Self-reflecting and self-monitoring can be used to build practice for change [17].

### 4.2. Strengths and Limitations of the Study

This study used MTM of behavior change, a new approach with high explanatory potential well supported in health promotion and public health research [43,44]. The findings of this study can pave the way for designing potential interventions based on MTM for helping college and university students quit gambling or participate in responsible gambling. The instrument used in this study demonstrated acceptable readability, validity, and reliability, suggesting its suitability for future cross-sectional and interventional studies. The findings indicate that the instrument effectively measured the intended constructs. This implies that researchers can confidently utilize the instrument in future studies to assess gambling behavior and related constructs among college and university students.

This study acknowledges certain limitations. Firstly, it was conducted solely at one large university in the southwestern region of the United States. Consequently, the generalizability of the findings to other universities or populations may be limited. It is advisable for future research to include a more diverse range of universities or expand the sample to enhance the external validity of the study’s conclusions. Secondly, the study’s reliance on self-reported information is a limitation, as it introduces potential biases such as recall bias, dishonesty, and acquiescence bias. Participants may have difficulty accurately recalling their gambling behavior or may provide socially desirable responses. However, it is important to note that self-reported data remain the primary method for collecting information on attitudes and behaviors related to health behavior. To address these limitations, efforts were made to ensure confidentiality and anonymity. Thirdly, the lack of test–retest reliability assessment prevented us from examining the instrument’s consistency over time. However, this limitation also presents an opportunity for future research to explore the stability of the instrument by conducting test–retest studies. Conducting such studies would enhance the robustness of the instrument and provide valuable insights into its stability and consistency over time. Fourthly, we asked the students about their past 30-day behavior, which is not necessarily representative of typical behavior. Fifthly, we used an incentive through participation in a random number draw, which has the potential of being considered gambling itself. This may have affected the results by increasing the likelihood of those engaged in gambling participating more. We used this measure to increase participation in our survey. Perhaps future researchers may either provide incentives to all participants or completely refrain from incentives. Sixthly, we used only one questionnaire in assessing the explanatory potential of MTM. Future researchers may employ a battery of instruments based on other theories so that comparative studies can be undertaken. Lastly, it is important to note that due to the cross-sectional design of the study, causal relationships could not be established. Cross-sectional studies provide a snapshot of data at a specific point in time, making it challenging to determine the direction of causality between variables. While this study provides valuable insights into the associations and predictors of gambling behavior among university students, caution must be exercised in drawing definitive conclusions about causality. Future research, utilizing longitudinal or experimental designs, would be valuable in investigating the temporal relationships and causal mechanisms underlying gambling behavior among this population. Further, we cannot say that participants’ gambling behavior changed due to the cross-sectional design of the study, but these MTM constructs may map on to protective factors. More research is likely needed.

## 5. Conclusions

This study made significant findings regarding the prevalence of gambling behavior among university students, with over one-third of the participants reporting recent gambling activity. These tendencies raise concerns about the potential risks of problem gambling and pathological gambling among this population. Importantly, this study is pioneering in its application of the contemporary theoretical framework of MTM to understand and explain gambling behavior among university students. The results highlight the promise of the novel MTM framework in elucidating factors associated with quitting gambling behavior among university students. These findings suggest that future researchers can leverage the MTM framework to develop targeted interventions aimed at assisting college and university students in quitting gambling or engaging in responsible gambling practices.

## Figures and Tables

**Table 1 healthcare-11-02151-t001:** Descriptive statistics of the demographic variables (n = 1474).

Variable	Characteristics	Mean ± SD	n (%)
Age	-	25.3 ± 7.7	-
Gender	Male	-	497 (33.72)
	Female	-	954 (64.72)
	Other	-	23 (1.56)
Race/Ethnicity	White	-	595 (40.37)
	Black	-	74 (5.02)
	Hispanic	-	326 (22.12)
	Asian American	-	236 (16.01)
	Native American	-	5 (0.34)
	Multi-racial	-	171 (11.60)
	Other	-	67 (4.55)
Class	Freshmen	-	164 (11.13)
	Sophomore	-	179 (12.14)
	Junior	-	318 (21.57)
	Senior	-	355 (24.08)
	Graduate	-	407 (27.61)
	Professional	-	51 (3.46)
GPA	Less than 1.99	-	14 (0.95)
	2.00–2.49	-	43 (2.92)
	2.50–2.99	-	133 (9.02)
	3.00–3.49	-	374 (25.37)
	3.50–4.00	-	910 (61.74)
Living condition	Off campus	-	1365 (92.61)
	On campus	-	109 (7.39)
Employment	Yes	-	1032 (70.01)
	No	-	442 (29.99)
Working hours per week *	-	26.4 ± 12.3	-
Self-reported gambling problem	Yes	-	64 (4.34)
	No	-	1410 (95.66)
Participated in gambling at least one time in the past month	Yes	-	562 (38.13)
	No	-	912 (61.87)
Type of gambling	Single type	-	791 (53.66)
	Multiple type	-	683 (46.34)

* Only 1032 (70.01%) students had employment.

**Table 2 healthcare-11-02151-t002:** Internal consistency, reliability, and convergent validity measurements for the different MTM constructs.

Subscale	Cronbach’s Alpha (95% Confidence Interval)	McDonald’s Omega	AVE
Perceived Advantage	0.93 (0.92, 0.93)	0.93	0.74
Perceived Disadvantage	0.88 (0.87, 0.89)	0.87	0.62
Behavioral Confidence	0.90 (0.89, 0.91)	0.91	0.66
Changes in the Physical Environment	0.77 (0.75, 0.79)	0.70	0.50
Overall Initiation Scale	0.80 (0.79, 0.82)	-	-
Emotional Transformation	0.92 (0.91, 0.93)	0.92	0.80
Practice for Change	0.83 (0.82, 0.85)	0.84	0.63
Changes in the Social Environment	0.80 (0.78, 0.82)	0.80	0.58
Overall Sustenance Scale	0.89 (0.88, 0.90)	-	-

**Table 3 healthcare-11-02151-t003:** Descriptive statistics of multi-theory model constructs of behavior change (n = 1474).

	Students Who Had Not Participated in Gambling in the Past Month (n = 912)			Students Who Had Participated in Gambling in the Past Month (n = 562)			
Constructs	Possible Score Range	Observed Score Range	Mean ± SD	Possible Score Range	Observed Score Range	Mean ± SD	*p*-Value
Initiation	0–4	0–4	3.79 ± 0.70	0–4	0–4	2.95 ± 1.37	<0.0001
Perceived Advantage (PA)	0–20	0–20	15.30 ± 5.37	0–20	0–20	11.23 ± 5.67	<0.0001
Perceived Disadvantage (PDA)	0–20	0–20	3.48 ± 4.43	0–20	0–20	4.63 ± 4.20	<0.0001
Participatory Dialogue (PA–PDA)	−20–+20	−9–+20	11.82 ± 6.99	−20–+20	−15–+20	6.60 ± 6.29	<0.0001
Behavioral confidence	0–20	0–20	17.11 ± 3.85	0–20	0–20	14.22 ± 4.87	<0.0001
Changes in the physical environment	0–12	0–12	7.40 ± 3.49	0–12	0–12	6.17 ± 3.63	<0.0001
Sustenance	0–4	0–4	3.17 ± 1.18	0–4	0–4	1.78 ± 1.48	<0.0001
Emotional transformation	0–12	0–12	10.87 ± 2.13	0–12	0–12	9.43 ± 3.04	<0.0001
Practice for change	0–12	0–12	10.20 ± 2.55	0–12	0–12	9.13 ± 3.08	<0.0001
Changes in the social environment	0–12	0–12	9.50 ± 3.06	0–12	0–12	8.68 ± 3.34	<0.0001

**Table 4 healthcare-11-02151-t004:** Hierarchical multiple regression predicts the likelihood of initiation for those who had not participated in gambling during the past month (n = 912).

Variables		Model 1		Model 2		Model 3		Model 4	
		β	*p*-Value	β	*p*-Value	β	*p*-Value	β	*p*-Value
Intercept		3.442	<0.0001	3.194	<0.0001	2.472	<0.0001	2.453	<0.0001
Type of Gambling	(Single type reference)	0.020	0.6857	0.050	0.2908	0.091	0.0453	0.097	0.032
Age		0.006	0.0892	0.004	0.2159	0.002	0.5327	0.002	0.5299
Sex	Female (Male reference)	0.138	0.0060	0.144	0.0032	0.090	0.0577	0.083	0.0776
Race/Ethnicity	Hispanic (White reference)	0.138	0.0253	0.147	0.0144	0.139	0.0158	0.124	0.0315
	Asian	0.023	0.7422	0.042	0.5353	0.057	0.3741	0.051	0.4305
	Others	−0.065	0.3001	−0.066	0.2772	−0.049	0.4039	−0.053	0.3595
Class	Sophomore + Junior (Freshmen reference)	0.030	0.6575	0.015	0.8166	0.009	0.8799	0.007	0.9043
	Senior	0.020	0.7637	0.007	0.9153	0.002	0.9752	0.001	0.9852
	Graduate	0.060	0.4099	0.058	0.4073	0.012	0.8549	0.016	0.8141
GPA	(Less than 3.50 reference)	0.071	0.1524	0.049	0.3101	0.057	0.2191	0.064	0.1654
Participatory dialogue				0.025	<0.0001	0.017	<0.0001	0.016	<0.0001
Behavioral confidence						0.052	<0.0001	0.047	<0.0001
Changes in the physical environment								0.015	0.0237
R^2^		0.029		0.088		0.160		0.165	
F		2.70	0.0029	7.90	<0.0001	14.28	<0.0001	13.63	<0.0001
ΔR^2^				0.059		0.072		0.005	

Adjusted R^2^ of model 4 = 0.153.

**Table 5 healthcare-11-02151-t005:** Hierarchical multiple regression predicting the likelihood of sustenance for those who had not participated in gambling during the past month (n = 912).

Variables		Model 1		Model 2		Model 3		Model 4	
		β	*p*-Value	β	*p*-Value	β	*p*-Value	β	*p*-Value
Intercept		3.078	<0.0001	0.816	0.0002	0.813	0.0001	0.789	0.0003
Type of Gambling	(Single type reference)	−0.571	<0.0001	−0.536	<0.0001	−0.513	<0.0001	−0.512	<0.0001
Age		−0.005	0.3984	−0.012	0.0227	−0.013	0.0102	−0.013	0.0113
Sex	Female (Male reference)	0.335	<0.0001	0.195	0.0081	0.154	0.0345	0.153	0.0348
Race/Ethnicity	Hispanic (White reference)	0.406	<0.0001	0.394	<0.0001	0.365	<0.0001	0.366	<0.0001
	Asian	0.340	0.0024	0.389	0.0001	0.372	0.0002	0.373	0.0002
	Others	0.062	0.5410	0.101	0.2673	0.089	0.3164	0.093	0.2992
Class	Sophomore + Junior (Freshmen reference)	−0.086	0.4292	−0.096	0.3224	−0.133	0.1637	−0.133	0.1643
	Senior	−0.004	0.9704	−0.043	0.6596	−0.046	0.6259	−0.049	0.6054
	Graduate	0.185	0.1133	0.050	0.6346	0.036	0.7257	0.033	0.7474
GPA	(Less than 3.50 reference)	0.050	0.5348	0.037	0.6066	0.014	0.8394	0.017	0.8133
Emotional transformation				0.235	<0.0001	0.135	<0.0001	0.134	<0.0001
Practice for change						0.115	<0.0001	0.110	<0.0001
Changes in the social environment								0.010	0.447
R^2^		0.108		0.278		0.306		0.306	
F		10.87	<0.0001	31.54	<0.0001	32.98	<0.0001	30.47	<0.0001
ΔR^2^				0.171		0.028		0.0004	

Adjusted R^2^ of model 4 = 0.296.

**Table 6 healthcare-11-02151-t006:** Hierarchical multiple regression predicting the likelihood of initiation for those who had participated in gambling during the past month (n = 562).

Variables		Model 1		Model 2		Model 3		Model 4	
		β	*p*-Value	β	*p*-Value	β	*p*-Value	β	*p*-Value
Intercept		3.336	<0.0001	2.749	<0.0001	1.595	<0.0001	1.563	<0.0001
Type of Gambling	(Single type reference)	−0.415	0.0002	−0.242	0.024	−0.120	0.2437	−0.102	0.3208
Age		−0.022	0.0043	−0.016	0.028	−0.017	0.0155	−0.017	0.0176
Sex	Female (Male reference)	0.493	<0.0001	0.399	0.0002	0.372	0.0002	0.352	0.0005
Race/Ethnicity	Hispanic (White reference)	0.099	0.5133	0.028	0.845	0.036	0.7879	0.001	0.9946
	Asian	0.089	0.5961	0.144	0.3674	0.196	0.194	0.160	0.2892
	Others	−0.170	0.2485	−0.144	0.2986	−0.080	0.5447	−0.091	0.4869
Class	Sophomore + Junior (Freshmen reference)	−0.111	0.5042	−0.179	0.2544	−0.177	0.2333	−0.169	0.254
	Senior	0.046	0.7752	−0.013	0.9306	0.020	0.8932	0.028	0.8476
	Graduate	−0.148	0.3866	−0.242	0.1347	−0.282	0.0666	−0.271	0.077
GPA	(Less than 3.50 reference)	0.329	0.0051	0.301	0.0065	0.283	0.007	0.269	0.0102
Participatory dialogue				0.071	<0.0001	0.054	<0.0001	0.052	<0.0001
Behavioral confidence						0.087	<0.0001	0.073	<0.0001
Changes in the physical environment								0.040	0.0137
R^2^		0.105		0.203		0.286		0.294	
F		6.43	<0.0001	12.71	<0.0001	18.34	<0.0001	17.56	<0.0001
ΔR^2^				0.098		0.084		0.008	

Adjusted R^2^ of model 4 = 0.277.

**Table 7 healthcare-11-02151-t007:** Hierarchical multiple regression predicting the likelihood of sustenance for those who had participated in gambling during the past month (n = 562).

Variables		Model 1		Model 2		Model 3		Model 4	
		β	*p*-Value	β	*p*-Value	β	*p*-Value	β	*p*-Value
Intercept		2.730	<0.0001	1.460	<0.0001	1.371	<0.0001	1.456	<0.0001
Type of Gambling	(Single type reference)	−0.653	<0.0001	−0.563	<0.0001	−0.557	<0.0001	−0.558	<0.0001
Age		−0.036	<0.0001	−0.033	<0.0001	−0.033	<0.0001	−0.034	<0.0001
Sex	Female (Male reference)	0.461	<0.0001	0.400	0.0004	0.393	0.0005	0.394	0.0005
Race/Ethnicity	Hispanic (White reference)	0.521	0.0010	0.576	0.0002	0.573	0.0002	0.565	0.0002
	Asian	0.480	0.0064	0.649	0.0002	0.634	0.0002	0.629	0.0002
	Others	0.177	0.2476	0.255	0.0833	0.256	0.0825	0.238	0.1065
Class	Sophomore + Junior (Freshmen reference)	−0.399	0.0214	−0.410	0.0137	−0.419	0.0117	−0.396	0.0175
	Senior	−0.187	0.2689	−0.227	0.1631	−0.212	0.1919	−0.203	0.2119
	Graduate	−0.286	0.1089	−0.312	0.0683	−0.335	0.0509	−0.322	0.06
GPA	(Less than 3.50 reference)	0.119	0.3293	0.037	0.7548	0.038	0.7468	0.042	0.7214
Emotional transformation				0.129	<0.0001	0.095	0.0007	0.104	0.0003
Practice for change						0.046	0.0949	0.060	0.0368
Changes in the social environment								−0.035	0.0903
R^2^		0.169		0.236		0.240		0.244	
F		11.23	<0.0001	15.45	<0.0001	14.45	<0.0001	13.60	<0.0001
ΔR^2^				0.067		0.004		0.004	

Adjusted R^2^ of model 4 = 0.226.

## Data Availability

The data presented in this study are available upon request from the corresponding author. The data are not publicly available due to the presence of ethical reasons.

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
