# Peer review of "Examining the Gambling Behavior of University Students: A Cross-Sectional Survey Applying the Multi-Theory Model (MTM) of Health Behavior Change in a Single Institution"

_healthcare, 2023, doi:10.3390/healthcare11152151_

Round 1

Reviewer 1 Report

In the current study, the authors set out to identify and explain gambling behavior among university students. Research examining factors involved in behavioral addiction and treatment is important, though the manuscript could be improved by addressing some concerns.

Overall:

-The authors are encouraged to thoroughly proofread the manuscript for errors in grammar and formatting (e.g., inconsistent use of spacing, not closing the parentheses in lines 55-58, and using “Hispanics” instead of Hispanic in line 188, capitalization of campus in line 279).

Introduction:

-The manuscript would benefit from the authors further explaining the theory behind the multi-theory model as well as the literature to date either supporting it or not supporting it in gambling or other problematic behaviors. It will be helpful for the reader to know how the model has been used and how this paper addresses gaps in the existing literature.

-It would be helpful for the authors to include a citation when defining pathological gambling as they did with problem gambling.

- The aim of the paper is sometimes unclear as the abstract identified the aims as “identify and explain gambling behavior among university students,” but the introduction identifies the questions guiding the study as determining “to what extent do the correlates…explain the initiation of quitting gambling behavior among university students while controlling for demographic covariates… [and] to what extent do the …explain the maintenance of quitting gambling behavior…”

Materials and Methods:

- If the authors’ intent was to examine initiation and maintenance of quitting gambling behavior, the sample doesn’t seem to be ideal for this purpose. Investigators know whether the student gambled in the last 30 days and used this in subsequent models. However this doesn’t take into account whether the gambling behavior was problematic or not and more importantly it doesn’t take into account whether the past month was typical behavior or whether a change was made. It’s also odd that students were asked if they found their gambling problematic and these findings were not reported anywhere in the manuscript.

-It would be helpful to include a citation when describing the survey. It’s unclear whether the survey has been previously validated or if an additional aim of the paper is validation of a new measure.

 -If the authors’ intent was to validate the survey, it would be helpful to include standard measures that target constructs included in the survey and statistically test this measure against them to thoroughly assess for construct validity.

Discussion:

-The manuscript would benefit from the authors discussing all significant findings and how these findings align with or do not align with specific findings in previous literature. Previous literature is briefly mentioned, but not discussed and changes in the social environment were not reviewed or discussed with initiation findings.

-It would be helpful to discuss ways the model factors could be built in the implications for practice section and remove this from other sections. Citations should be included.

-The manuscript would be improved by tempering language around how this study’s findings may “provide ways to design an effective intervention…” These findings may improve a knowledge base related to behavior that may inform future interventions, but there was no intervention used or tested in this study and a population with clear difficulties with gambling behavior wasn’t used.

-It may be an overstatement to say that the survey “produced consistent and trustworthy results” when it was only tested at one timepoint. Authors go on to mention this as a limitation and acknowledge that stability wasn’t examined.

Author Response

We thank you for sharing your time and your kind comments to strengthen our paper. We have prepared a point by point response document.  Please see the attachment.

Reviewer 2 Report

The paper addresses the important topic of gambling behaviour among students. The authors approach the issue using the Multi-Theory Model, which has been used in studies of many other factors related to problematic health behaviour, which is new to the paper.

The introduction is written correctly. The problem of gambling behaviour is presented: its correlates and consequences and the main assumptions of the Multi-Theory Model. The authors also state the purpose of the research and pose two research questions derived from the assumptions of the MTM.

The research procedure is conducted similarly to other texts applying the MTM. The recruitment of participants was multistage. An interesting feature is the incentive through participation in a lottery, which otherwise could be treated as gambling itself. 

The time span (one month) as well as the number of events (at least one) differentiating the research population into gamblers and non-gamblers may be questionable.  The characteristics of compulsive gamblers are certainly different from the population of occasional gamblers. Certainly, this factor may have an impact on the research results achieved. 

The results of the study are presented in tabular form. They are readable and clear. Statistical analyses are carried out correctly. The presented models extended with additional explanatory variables provide valuable research material.

The discussion of the research results is conducted in a transparent manner. What I find somewhat lacking is the cross-referencing of research results using MTM to research based on the assumptions of other theories, which is the essence of the discussion....

Implications for practice are certainly valuable.

In addition, the authors are aware of the limitations of their research (one institution, among others). The value of the text is certainly the validation of the questionnaire and its application to a new research area. I think it would have been good to add as an appendix the questionnaire itself (I understand that it can be found elsewhere on the internet). Nevertheless, it would make it easier for the reader to understand the text

Author Response

(The authors gave the same response as above.)

Reviewer 3 Report

In their manuscript entitled “Examining the Gambling Behavior of University Students: A Cross-Sectional Survey Applying the Multi-Theory Model 3 (MTM) of Health Behavior Change in a Single Institution”, the Authors present a cross-sectional study in which they tested whether and how a novel fourth-generation multi-theory model (MTM) of health behavior change explains the initiation and maintenance of the quitting gambling behavior among university students.

This is a very interesting and tamily study. It seems well designed, well conducted and well written in most of its parts. However, in my opinion, there are some minor revisions to be fixed.

1)      In the Introduction section, I suggest Authors to add some more details about the MTM model. In particular, I suggest to integrate the section with evidence, if present, about the efficacy of this model to explain other mental conditions and disorders. Are there other articles that used this model that can be cite? It would be interesting to have in the introduction a brief overview of the literature about the model and it’s use over time.

2)      The Authors provided a critical analysis of their work in the ”Strengths and Limitations of the Study” section, pointing out the limits linked to the use of a self-report instrument. Among these limits, I suggest Authors to add also the limit of using one single questionnaire.  I understand that the objective of the Authors was to specifically test the MTM model that consists of just a 39-item survey. However, measuring complex construct, such as “gambling”, by using one self-report questionnaire could provide a partial assessment, compared to evaluations that consider a battery of tests. I think that this point should be discussed at least in the ”Strengths and Limitations of the Study” section.  

Author Response

(The authors gave the same response as above.)

Round 2

Reviewer 1 Report

In the current study, the authors set out to identify and explain gambling behavior among university students. Research examining factors involved in behavioral addiction and treatment is important, and the authors have made many of the requested revisions. However, the manuscript could be improved by addressing additional concerns.

Introduction:

-The authors did add more information about the multi-theory model. However, it may still be helpful to identify whether the model is supported in the studies examining the health behaviors the authors mentioned (e.g., vaping, smoking, substance use). Generally more discussion of how the model has been used and supported or not supported in previous literature will be helpful.

- The current primary aim of the paper cannot be fully explored as the authors can’t identify those that changed and those that did not change their gambling behavior. Authors can examine what is related to gambling or not gambling in the last 30 days, but cannot accurately say they are examining behavior change.

-The manuscript may benefit from removal of the secondary aim. This may be worthy of a separate manuscript.

Materials and Methods:

-As mentioned previously, the manuscript would benefit from removing sections that attempt to validate the survey instrument.

Discussion:

-Because the authors cannot say that participants’ gambling behavior changed, it might be helpful to mention how these constructs may also map on to protective factors. More research is likely needed.

N/A

Author Response

Dear reviewer,

Thank you very much for your time and kind comments to strengthen our paper. We have address your comments. Please see the attachment.

Thank you,

Best regards,

Sidath
